# High Seebeck Coefficient from Screen-Printed Colloidal PbSe Nanocrystals Thin Film

**DOI:** 10.3390/ma15248805

**Published:** 2022-12-09

**Authors:** Viviana Sousa, Guillaume Savelli, Oleg I. Lebedev, Kirill Kovnir, José H. Correia, Eliana M. F. Vieira, Pedro Alpuim, Yury V. Kolen’ko

**Affiliations:** 1Center of Physics of the Universities of Minho and Porto, University of Minho, 4710-057 Braga, Portugal; 2International Iberian Nanotechnology Laboratory, 4715-330 Braga, Portugal; 3University Grenoble Alpes, CEA-Liten, 17 av. Des Martyrs, 38000 Grenoble, France; 4Laboratoire CRISMAT, UMR 6508, CNRS-ENSICAEN, 14050 Caen, France; 5Department of Chemistry, Iowa State University, Ames, IA 50011, USA; 6Ames National Laboratory, U.S. Department of Energy, Ames, IA 50011, USA; 7CMEMS-UMinho, University of Minho, 4800-058 Guimarães, Portugal; 8LABBELS–Associate Laboratory, 4710-057 Braga, Portugal

**Keywords:** nanomaterials, heating-up synthesis, semiconductor ink, chalcogenides, microstructure, Seebeck coefficient

## Abstract

Thin-film thermoelectrics (TEs) with a thickness of a few microns present an attractive opportunity to power the internet of things (IoT). Here, we propose screen printing as an industry-relevant technology to fabricate TE thin films from colloidal PbSe quantum dots (QDs). Monodisperse 13 nm-sized PbSe QDs with spherical morphology were synthesized through a straightforward heating-up method. The cubic-phase PbSe QDs with homogeneous chemical composition allowed the formulation of a novel ink to fabricate 2 μm-thick thin films through robust screen printing followed by rapid annealing. A maximum Seebeck coefficient of 561 μV K^−1^ was obtained at 143 °C and the highest electrical conductivity of 123 S m^−1^ was reached at 197 °C. Power factor calculations resulted in a maximum value of 2.47 × 10^−5^ W m^−1^ K^−2^ at 143 °C. To the best of our knowledge, the observed Seebeck coefficient value is the highest reported for TE thin films fabricated by screen printing. Thus, this study highlights that increased Seebeck coefficients can be obtained by using QD building blocks owing to quantum confinement.

## 1. Introduction

Thin-film thermoelectric (TE) generators have recently attracted considerable attention in the information and communication technology applications, particularly for the long-term powering of internet-of-things (IoT) devices [1]. Classically, thin-film TEs are fabricated by means of effective but expensive vacuum-based techniques, such as chemical or physical vapor deposition techniques [2,3,4]. One key challenge in fabricating thin-film TEs is establishing alternative cheap deposition methods that yield high-quality thin film of TE material over large-area substrate [5,6,7]. For instance, the ability to use solution-based screen printing to deposit TE thin films would be advantageous owing to the cost-effectiveness and high throughput of this industrially relevant deposition technique [8].

Screen printing relies on a screen with a prepatterned mesh of specific size, where viscous ink is forced to pass through the mesh toward the substrate, with the help of a squeegee. This technique affords few-micrometer-sized TE thin films with compact material packing. Moreover, screen printing allows different substrates and patterns, which widens the range of TE generator geometries [9]. Notably, to improve ink adhesion to the substrate as well as its processability, the ink formulation demands an organic binder, which typically decreases the electrical conductivity of the resultant TE thin film. Accordingly, subsequent heat or chemical treatments of the screen-printed TE thin films are needed to achieve the required high electrical conductivity of the resultant functional film.

From the TE point of view, the intrinsically small dimensions of quantum dots (QDs) provide a natural reduction in thermal conductivity down to 0.9 W m^−1^ K^−1^ via phonon scattering at the QD interfaces [10]. Moreover, quantum confinement has been proposed to enable materials with outstanding Seebeck coefficients because of sharp features of carrier density-of-states near Fermi energy, in combination with selective carrier filtering at the QD interfaces [11,12,13]. Interestingly, QDs with TE properties could be used as building blocks for the fabrication of the TE thin films [14,15]. The QDs are typically synthesized by colloidal synthesis that employ long-chain organic capping ligands to passivate the QD surfaces and to provide a barrier to coalescence. Accordingly, the as-synthesized QDs are well-dispersible in non-polar organic solvents forming a stable colloidal dispersion, and hence, could be easily introduced into the formulation of the ink for screen printing.

One fundamental challenge in the TE thin film design involves the printed fabrication of the film from TE-relevant QDs. There are several hurdles in this direction during thin-film printing and processing, including the preservation of the QD particle sizes needed to exhibit the quantum confinement effect while avoiding any possible changes in the phase and chemical compositions of constituting QDs. Hence, screen-printed TE thin films from QDs are not common, and most of the reported works are dedicated to the screen printing of TE thin films from non-quantum materials, such as Bi_2_Te_3_ [1,16,17,18,19,20,21,22,23,24], Sb_2_Te_3_ [25,26], Ag_2_Se [27,28,29] or PbTe [30], obtained by high-temperature melting and high-energy ball milling. In this work, we synthesized crystalline and compositionally homogeneous colloidal PbSe QDs using a heating-up method. Stable and viscous ink based on PbSe QDs was then formulated using ethyl cellulose as a binder and terpineol as a solvent. This novel ink enables the reliable fabrication of PbSe thin films by means of screen printing followed by a rapid heat treatment. Finally, the temperature-dependent TE properties of the resultant ca. 2 μm-thick films consisting of PbSe QDs were investigated, and the observed electrical conductivity and Seebeck coefficient value trends are highlighted.

## 2. Experimental Methods

### 2.1. Materials

Lead (II) acetate trihydrate (Pb(ac)_2_∙3H_2_O, ≥99.99%), selenium (Se, 100 mesh powder, ≥99.5%), trioctylphosphine (TOP, 97%), oleic acid (OA, 90%), 1-octadecene (ODE, 90%), ethyl cellulose (EC, 48% ethoxyl) and terpineol were purchased from Sigma-Aldrich, St. Louis, MO, USA. Analytical reagent grade hexane and toluene (99.8%) were purchased from Fisher Scientific, Waltham, MA, USA. Acetone (≥99.5%), isopropanol (≥99.8%), and ethanol (≥99.8%) were purchased from Honeywell, Charlotte, NC, USA.

### 2.2. Synthesis of PbSe QDs

To prepare the PbSe QDs, a previously reported colloidal synthesis protocol was adapted and modified [31]. First, 1 M Se solution in TOP (TOP:Se) was prepared by dissolving 7.89 g of Se powder in 100 mL of TOP while stirring overnight at 60 °C. In a typical experiment to prepare PbSe QDs, Pb(ac)_2_∙3H_2_O (2.8 g, 7.4 mmol), OA (6.6 g, 23.4 mmol), ODE (50 mL, 156.3 mmol) and 1 M TOP:Se solution (12 mL) were loaded under ambient conditions into a 250 mL three-neck round-bottom flask, equipped with a magnetic stir bar, thermocouple, condenser, and vacuum adapter. Next, the flask was attached to a Schlenk line, and the mixture was degassed under vacuum at 130 °C for 2 h while stirring to remove low boiling liquids, such as possible water and acetic acid admixtures. Then, the vacuum was switched to Ar, and the water reflux was initiated. Under continuous stirring, the flask was then rapidly heated to 190 °C and kept at this temperature for 10 min. Next, the resultant dark reaction mixture was cooled to room temperature (RT) using a water bath. Afterwards, the as-synthesized PbSe QDs were washed by the addition of 25 mL of ethanol and collected by centrifugation at 9000 rpm for 5 min. The resultant solid was washed with solvent mixture of hexane and isopropanol (1:3) and again collected by centrifugation. Finally, the size of the PbSe QDs was selected by dispersing of centrifuged solid in 15 mL of toluene followed by centrifugation at 3000 rpm for 10 min. The resultant supernatant with colloidal PbSe QDs was stored in closed vial at 4 °C. The concentration of PbSe QDs in toluene was estimated gravimetrically, and adjusted to be 30 mg/mL.

### 2.3. Fabrication of PbSe QD Thin Films

#### 2.3.1. Ink Formulation

A 5% solution of ethyl cellulose in terpineol was prepared by dissolving the appropriate amount of EC in terpineol at 50 °C while stirring. To prepare an ink, 25% weight content of the PbSe QDs in solid form was added to the appropriate amount of 5% EC solution in terpineol. The ink was mixed first with a spatula and then probe sonication was used to improve QDs’ dispersion. The ink was sonicated on 5 min cycles with 5 s of sonication at 40% amplitude and 5 s pause. To achieve high dispersion of the PbSe QDs in the ink, 3 cycles were performed. Finally, the ink was let stir at 50 °C overnight.

#### 2.3.2. Substrate Preparation

A 500 nm-thick silicon dioxide (SiO_2_) layer was grown by plasma-enhanced chemical vapor deposition on Si (100) substrate. The resultant SiO_2_/Si substrate was then diced with 0.7 × 2 cm^2^ and 2 × 2 cm^2^ dimensions using an automatic dicing saw. The substrates were consecutively cleaned with acetone and isopropanol by ultrasonication for 5 min. Then, the substrates were dried by N_2_ flow, and finally, subjected to oxygen plasma cleaning for 10 min.

#### 2.3.3. Screen Printing

To obtain the TE thin films from PbSe QDs on SiO_2_/Si substrates, six layers of ink were screen-printed using a homemade screen printer having 180 threads cm^−1^ mesh count while drying at 80 °C for 3 min on a hot plate in between each deposition.

#### 2.3.4. Heat Treatment

The heat treatment was performed on a Thermal CVD MicroSys 400 System (Roth & Rau, Hohenstein-Ernstthal, Germany). The as-printed TE thin films were first annealed at 600 °C for 15 min under vacuum, and then under H_2_ gas flow of 100 sccm at 600 °C for 15 min to remove residual organics (ink components and QDs’ capping ligand).

### 2.4. Characterization

The phase composition of the PbSe QDs and the resultant thin films were analyzed using X-ray diffraction (XRD). The data were collected using an X’Pert PRO diffractometer (PANalytical, Malvern, UK) with Ni-filtered Cu *K*_α_ radiation (λ = 1.541874 Å) and a PIXcel detector. The XRD patterns were matched to International Centre for Diffraction Data (ICDD) PDF-4 database using the HighScore software package (PANalytical, Malvern, UK).

To investigate the morphology and the chemical composition of the synthesized PbSe QDs, high-angle annular dark-field scanning transmission electron microscopy (HAADF–STEM) and energy-dispersive X-ray spectroscopy in STEM mode (STEM−EDX) were carried out using a JEM-ARM200F electron microscope (Jeol, Tokyo, Japan), operated at 200 kV. QDs’ size distribution was estimated using ImageJ software package. The thickness and the surface morphology of the TE thin films were analyzed by scanning electron microscopy (SEM) using Quanta 650 FEG ESEM (FEI, Hillsboro, OR, USA).

Electrical resistivity and charge mobility were measured by a simultaneous Hall Effect and Van der Paw method at RT. These measurements allowed to determine the surface and volume concentration of carriers, electron mobility and electrical conductivity. The results obtained are an average of five measurements with 5% error.

The Seebeck coefficient and electrical conductivity, in the temperature range from RT to 200 °C, were measured using a ZEM-3 system (ULVAC). The scheme of measurement is available in the Appendix A. The results are an average of three measurements at four different temperatures (30 °C, 90 °C, 140 °C, and 200 °C) with three different thermal gradients (10 °C, 20 °C, and 30 °C). The results obtained are an average value with 7% error.

## 3. Results

### 3.1. Monodisperse, Phase-Pure and Chemically Uniform Cubic-Phase PbSe QDs

PbSe QDs were prepared by a facile heating-up colloidal synthesis, relaying on the reaction between organometallic Pb precursor and Se solution in TOP under inert atmosphere. The as-synthesized PbSe QDs adopted a nearly spherical shape with monodisperse size distribution of 13 ± 1 nm, as shown in the representative low-magnification and high-resolution HAADF–STEM images (Figure 1a,c,d). The XRD pattern of the as-synthesized PbSe QDs (Figure 2) matched well the standard pattern for cubic-phase PbSe (ICDD no. 04-002-6293). The square-like atomic arrangement and well-resolved lattice spacing, observed in the high-resolution HAADF–STEM image (Figure 1d), further confirm the formation of crystalline cubic-phase PbSe QDs. The STEM–EDX element maps evidence the uniform distribution of Pb and Se elements within the QDs (Figure 1b). To evaluate the thermal behavior of the as-synthesized PbSe QDs, the powdered sample was subjected to a thermogravimetric analysis in Ar, which indicates that the PbSe phase itself did not show any significant weight loss before 650 °C (Appendix A). At the same time, the observed ca. 6.3% of weight loss before 450 °C (Appendix A) was associated with the evaporation of the residual toluene solvent as well as the degradation of the oleate capping ligand [32].

### 3.2. Phase-Pure and Relatively Compact PbSe QD Thin Films

Having high-quality PbSe QDs in hand, we further formulated a novel ink for screen printing. Since oleate-capped colloidal QDs could be well-dispersed in non-polar organic solvents, the ink was formulated based on terpineol solvent while using ethyl cellulose as a binder [33]. This newly developed ink affords reliable screen printing of the PbSe QD thin films onto SiO_2_/Si rigid substrates under ambient conditions. Rapid annealing of the as-printed films in vacuum (15 min) and H_2_ (15 min) at a temperature of 600 °C was then conducted to largely eliminate the organic matter from the printed thin films to improve their electrical conductivity.

Notably, the heat treatment did not affect the morphology or phase compositions of the original PbSe QD building blocks. Specifically, the XRD analysis (Figure 2) of the as-fabricated thin film demonstrated the same crystal structure of cubic-phase PbSe (ICDD no. 04-002-6293). Moreover, additional reflections could be observed in the XRD pattern of the film (Figure 2), which are associated with the SiO_2_/Si from the film substrate. Regardless of the conducted heat treatment, the presence of the residual carbon was still detected in the final PbSe QD thin films, as confirmed by Raman spectroscopy (Appendix A) [34].

The morphology of the films was observed by SEM, and the results indicate that the surface of the film exhibited a nanoparticulate appearance (Figure 3a). This suggests the presence of pristine PbSe QDs in the film, i.e., the PbSe QDs were not molten during rapid heat treatment at 600 °C. Figure 3b displays a cross-sectional SEM image of a PbSe QD thin film, wherein a PbSe QDs layer of around 2 μm was observed. Regardless of the relatively compact appearance of the PbSe QD thin film, it could be observed from Figure 3 that the film exhibited surface roughness, as well as a certain degree of porosity. The latter was most likely developed because of the liberation of the organic matter during rapid heat treatment.

### 3.3. PbSe QD Thin Films with Low Electrical Conductivity but High Seebeck Coefficient

To explore a potential application of the fabricated nanostructured PbSe QD thin films as thermoelectrics, we investigated their electronic properties as well as temperature-dependent electrical conductivity (*σ*) and Seebeck coefficient (*S*). Electrical properties, measured at RT by the Hall effect, revealed that PbSe QD thin films exhibited a bulk carrier concentration of 3.8 × 10^18^ cm^−3^, electron mobility of 7.9 × 10^−1^ cm^2^ V^−1^ s^−1^, and a quite low *σ* = 50 S m^−1^. The *σ* and *S* were then measured from RT to 200 °C in air (Figure 4a). Importantly, no oxidation of the PbSe phase was detected during such measurements (Appendix A), highlighting a good stability of the PbSe QD thin films within this temperature range at ambient conditions. The *σ* increased with the increase in the temperature, showing typical semiconductor behavior, and accordingly, the highest *σ* = 123 S m^−1^ was reached at 200 °C. The *σ* value is expected to continue increasing with temperature. A maximum value of *S* = 561 μV K^−1^ was observed at 143 °C. The measured positive *S* values indicate that PbSe QDs are *p*-type semiconductors with holes as the majority carriers. Power factor (*PF*) calculations resulted in a maximum value of *PF* = 2.47 × 10^−5^ W m^−1^ K^−2^ at 143 °C (Figure 4b).

Notably, the thermoelectric characterization of the thin films is a non-trivial task. In the current study, the *σ* and *S* was also evaluated by using a two-probe custom-made equipment, and compared with the values above in Appendix A, obtained using the four-probe method.

## 4. Discussion

Phase-pure colloidal PbSe QDs with a monodisperse size of 13 nm and uniform chemical composition were obtained using the heat-up method. The QDs were formulated into novel ink and screen-printed on SiO_2_/Si substrate. After rapid heat treatment under an inert atmosphere, 2 μm-thick PbSe QD thin films were fabricated and their TE properties were investigated. To compare our results for the as-fabricated thin films, an extensive list of the reported screen-printed film thermoelectrics was analyzed and compared in Figure 5. Notably, screen-printable thin films reported so far typically employ TE particles obtained by high-temperature melting and high-energy ball-milling techniques for the fabrication of the films with a thickness of tens of microns [1,16,17,18,19,20,21,22,23,24,25,26,27,28,29,30,35,36,37,38,39]. In contrast, here, we show that it is possible to obtain 2 μm screen-printed PbSe QD thin films with interesting thermoelectric properties from QD building blocks obtained by the colloidal synthesis route.

The *S* at room temperature for our PbSe thin films reached 176 μV K^−1^ and a maximum of 561 μV K^−1^ at 143 °C, outperforming reported screen-printed TE thin films (Figure 5). For instance, Pb_0.98_Na_0.02_Te–2 mol% SrTe screen-printed thin film only reached a maximum of 400 μV K^−1^ at 350 °C [30]. The observed high *S* in our thin films is most likely related to the quantum confinement effect of key constituent PbSe particles. Notably, our PbSe nanocrystals had an average size of 13 nm (Figure 1), which was well below the quantum confinement upper limit of 46 nm set by the Bohr radius of an exciton (electron-hole pair) for PbSe QDs [12]. In turn, quantum confinement has been theoretically proposed to enable materials with an outstanding Seebeck coefficient [40,41], and several studies have experimentally verified this prediction in particular conditions [42,43,44]. Specifically, the Seebeck voltage arises from the disorder of the equilibrium distribution of charge carriers, described by the Fermi–Dirac distribution. It is a measure of the variation in *σ*(*E*) above and below the Fermi surface, i.e., by the logarithmic derivative of *σ* with *E*. From another point of view, *S* can also be understood as the average energy per carrier. Outstanding *S* can be reached by quantum confinement via (i) inducing sharp features in the carrier’s density of electronic states near Fermi energy (high *dn*(*E*)/*dE*), and (ii) introducing energy barriers to selectively scatter charge carriers according to their sign and energy (enhancing *dμ*(*E*)/*dE*).

In the literature, thin films produced from different lead chalcogenide QDs showed a higher Seebeck coefficient than the respective bulk material. For example, Yan and co-workers [45] produced thin films from PbTe nanowires with diameters below 30 nm reaching high *S* > 470 μV K^−1^, while Tai and co-workers [46] obtained *S* = 628 μV K^−1^ for their thin films, which are a huge enhancement in comparison with *S* = 265 μV K^−1^ of bulk PbTe. In another study, Zhou and co-workers [47] reported thin films from PbTe QDs of 8–14 nm, which reached *S* above 500 μV K^−1^. Yang and co-workers [48] fabricated thin films of PbSe(SnS_2_) QDs with *S* = –420 μV K^−1^ near 200 °C. A higher Seebeck coefficient value of 580 μV K^−1^ at room temperature was also reported by Nugraha and co-workers [49] for PbS QD thin films. Interestingly, Yang and co-workers [50] prepared PbSe QD thin films exhibiting a Seebeck coefficient > 600 μV K^−1^, which is considerably high when compared with the best *S* = 380 μV K^−1^ near 100 °C for bulk PbSe [51].

The observed decreasing trend in the *S* after reaching its maximum at 143 °C (Figure 4a) could be due to the excitation of the minority charge carriers over the bandgap (*E_g_*), despite the fact that some technical issues with the measurement cannot be completely ruled out as a reason for a sharp six-fold decrease in the *S* value at 200 °C. Interestingly, regardless of the observed drop in thermopower at 200 °C, the electrical conductivity kept increasing with temperature (Figure 4a). Such behavior could be rationalized by the different compensation effect occurring due to the electrons from valence and conduction bands, that does not affect the electrical conductivity and the thermopower in a similar manner. Specifically, the thermopower is quite sensitive to the band asymmetry in relation to the Fermi level. As the temperature increases, electronic states in a thicker shell around the Fermi level will participate in the thermoelectric transport. Accordingly, the change in the Seebeck coefficient with temperature may not be uniform [52]. The bandgap of the produced material can be estimated following Goldsmid–Sharp formalism as *E_g_* = 2*e*|*S_max_*|*T_max_* = 0.50 eV, where *e* is the elementary charge, *S_max_* is the maximum Seebeck value reached, and *T_max_* is the absolute temperature at which *S_max_* occurs [53]. The bandgap calculated for the as-fabricated PbSe QD thin films (*E_g_* = 0.50 eV) is larger than the one of bulk PbSe (*E_g_* = 0.27 eV) [16]. The observed increased bandgap of the thin films might be due to the quantum confinement effect of the PbSe QDs with small particle size (Figure 1) [54].

The *σ* of PbSe QD thin films, near RT, measured by the Hall effect was quite low *σ* = 50 S m^−1^. Even at 200 °C, the maximum value obtained was *σ* = 123 S m^−1^, which is significantly lower when compared with *σ* = 1.84 × 10^4^ S m^−1^ of bulk PbSe [51], or with other screen-printed thin-film TE materials presented in Figure 5. We then increased the thickness of our PbSe QD thin films from 2 μm to 3.5 μm by decreasing the screen printer’s mesh counts from 180 to 120 threads cm^−1^, and determined the *σ* at RT. The *σ* results listed in Appendix A confirm that a thicker PbSe QD layer could increase the *σ* of the film by nearly twice as much [20]. Nevertheless, the obtained value *σ* = 165 S m^−1^ was still far from the 10^4^ S m^−1^ order when compared with the other screen-printed materials (Figure 5).

On the basis of the above results, one could deduce that although we were able to develop a baseline approach to PbSe QD thin films with an excellent Seebeck coefficient, the electrical conductivity of the thin films was prohibitively low with respect to real TE applications. The measured low value of the *σ* can be related to the small contact conductance between the spherical semiconducting QDs [55], since it is known that conventional semiconducting behavior will not be achieved if the spherical QDs are not fused into bulk by heat treatment [56]. Besides sintering, another interesting opportunity to increase the conductivity of the network of spherical semiconductor QDs is afforded by conformal deposition of additional conductive material onto connected spherical QDs, for example, using atomic layer deposition, thus increasing the electrical conductivity through the increase in contact area between spherical QDs [55,57]. One could also synthesize faceted QDs, for example, cubic-shaped, and then face-to-face assemble them into the QD thin film. Such a film is expected to exhibit enhanced *σ* as a consequence of the significantly increased contact area between cubic-shaped QDs as compared to spherical-shaped QDs. The explorations of the aforementioned approaches towards the increase in the *σ* of our PbSe QD thin films are the subject of our ongoing research efforts.

## 5. Conclusions

We demonstrate screen printing technology as an easy-to-scale-up approach to fabricate thermoelectric PbSe thin films with a thickness of a few micrometers and potential for near-RT application. This was achieved through (i) a straightforward colloidal synthesis of high-quality PbSe QD building blocks by means of the heat-up method, which allowed obtaining narrow-size and spherical-shape PbSe QDs, (ii) judicious formulation of a novel screen-printable PbSe QD ink obtained from well-dispersed QDs into a polymeric matrix, and (iii) a practical fabrication of the PbSe QD thin films by means of screen printing followed by rapid annealing, applied to eliminate ink organic precursors that could hinder QDs’ charge transport. We propose that owing to the quantum confinement effect from the PbSe QDs’ small size, the resultant thin films exhibit an outstanding Seebeck coefficient (561 μV K^−1^ at 143 °C), revealing the high potential of PbSe QDs for TE applications near RT. At the same time, the intrinsic shortcoming of the QD building blocks, namely, the small contact conductance between the spherically shaped QDs, led to very poor electrical conductivity (123 S m^−1^) of the final PbSe QD thin films. Studies on overcoming this shortcoming with the help of atomic layer deposition of the conductive layer are currently underway in our laboratories.

## Figures and Tables

**Figure 1 materials-15-08805-f001:**
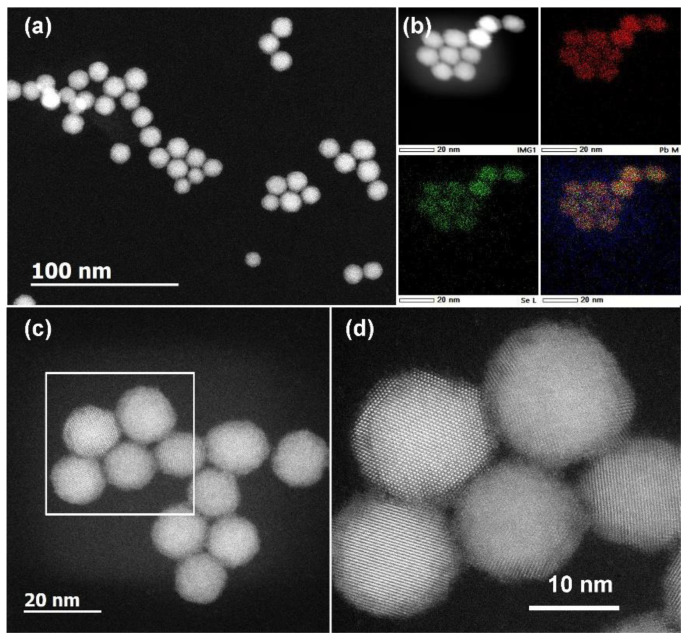
Representative low-magnification (**a**,**c**) and high-resolution (**d**) HAADF–STEM images of the as-synthesized PbSe QDs, together with the simultaneously collected STEM–EDX elemental maps of Pb, Se, and their mixture (**b**).

**Figure 2 materials-15-08805-f002:**
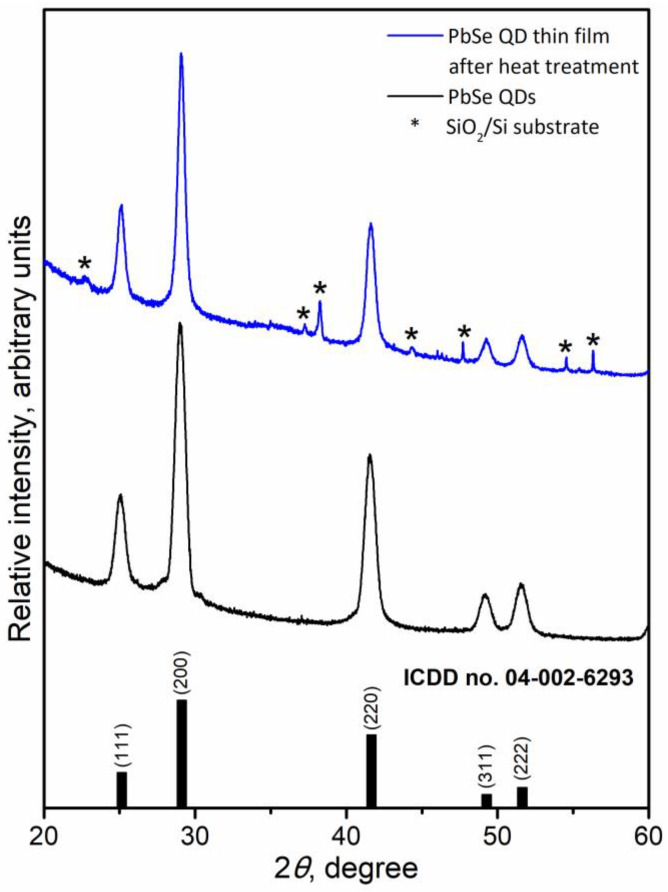
XRD patterns of the as-synthesized PbSe QDs (bottom) and the resultant PbSe QD thin film (top).

**Figure 3 materials-15-08805-f003:**
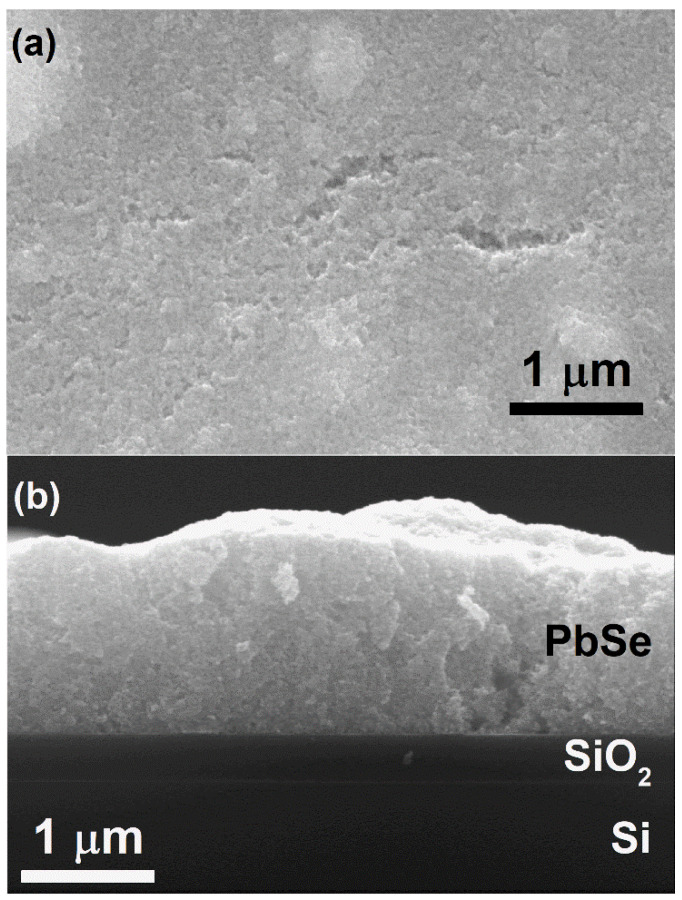
Representative surface (**a**) and cross-sectional (**b**) SEM images of the PbSe QD thin film, fabricated through screen printing followed by rapid heat treatment.

**Figure 4 materials-15-08805-f004:**
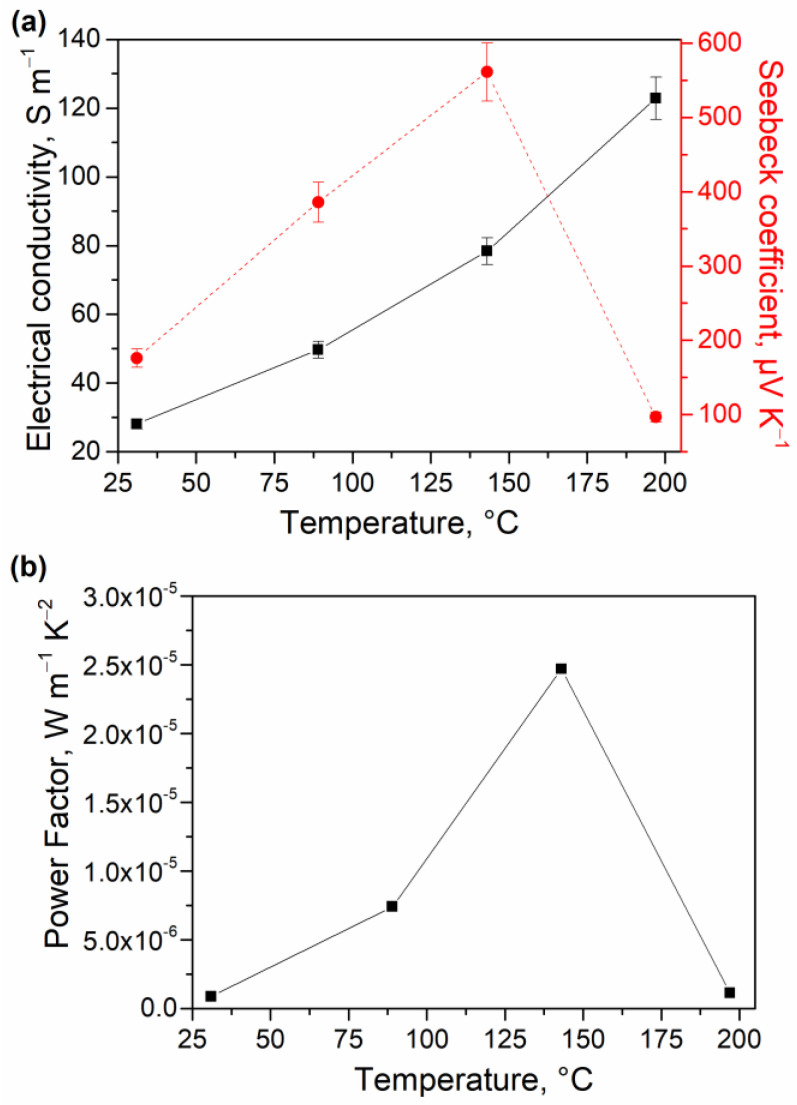
Temperature-dependent electrical conductivity and Seebeck coefficient (**a**), as well as Power Factor (**b**) of the *p*-type PbSe QD thin films.

**Figure 5 materials-15-08805-f005:**
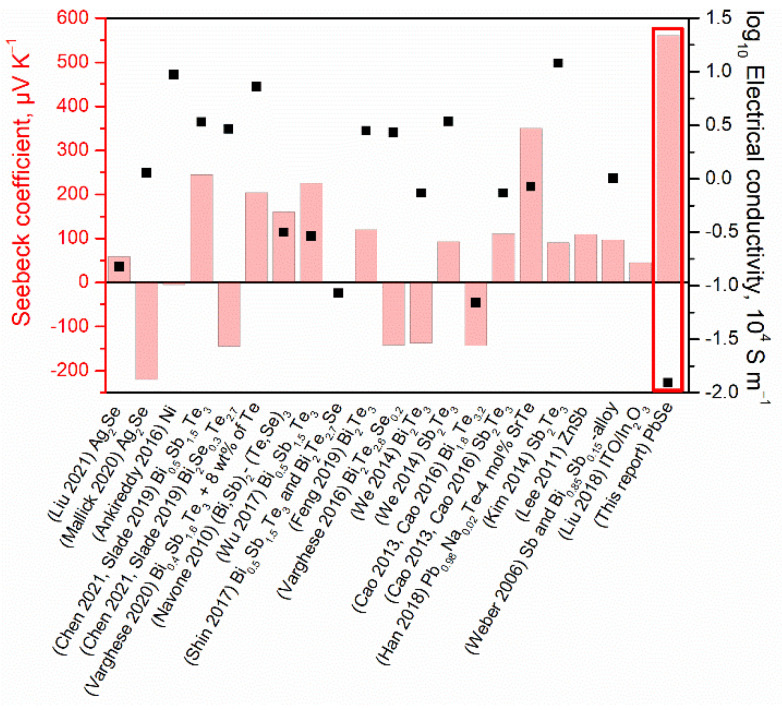
Comparison of the Seebeck coefficient and electrical conductivity for different reported thermoelectric thin films fabricated using the screen printing technique [1,16,17,18,19,20,21,22,23,24,25,26,27,28,29,30,35,36,37,38,39].

## Data Availability

Not applicable.

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
