# Peer review of "High Seebeck Coefficient from Screen-Printed Colloidal PbSe Nanocrystals Thin Film"

_materials, 2022, doi:10.3390/ma15248805_

Round 1

Reviewer 1 Report

As seen in figure 4, the Seebeck and electrical conductivity increase with temperature up to 143 C. This trend is a little out of the general trends, as electrical conductivity varies directly with concentration, while S varies inversely with temperature. Why this trend? What is the reason that S increases with temperature up to 143 C? Also, after the 143 C, if there is an excitation of the minority charge carriers, that hampers the Seebeck value. If that so, Is that affect electrical conductivity? As electrical conductivity is also affected by the recombination led by minority carriers.

Reviewer 2 Report

Through the review, I think this is a very nicely written paper, and the author analyzed and discussed the results in detail and correctly. The analysis developed in this paper is correct and the obtained results are interesting. The paper has sufficient novelty and covers the scope of the Materials journal. Therefore, the manuscript may consider for publication in the Materials journal after responding to the following comments and revising the manuscript properly.

1. The novelty of the work is missing in the introduction section. Explain it properly.

2. Improve language throughout the manuscript.

3. Improve the introduction based on the following reference on optical and nanomaterials: (Chapter 2 - Classification and properties of nanoparticles, Book: Nanoparticle-Based Polymer Composites, 2022, Pages 15-54; ACS Appl. Electron. Mater. 2021, 3, 9, 3715–3746)

Reviewer 3 Report

In this article the authors propose screen printing as a  technology to fabricate TE thin films from colloidal PbSe quantum dots (QDs).

·       In the introduction, the authors should clearly highlight the novelty of their work in comparison to previous works.

·       The conclusion section should be extended and improved to help a reader not familiar with the topic.

·       The authors should use arbitrary units instead of a. u. in the label of Figure 2.

Round 2

Reviewer 1 Report

Questions no.2 is still not properly responded 

Also, after the 143 °C, if there is an excitation of the minority charge carriers, that hampers the Seebeck value. If that so, is that affect electrical conductivity? As electrical conductivity is also affected by the recombination led by minority carriers.
